# Weight Loss and Competition Weight in Ultimate Fighting Championship (UFC) Athletes

**DOI:** 10.3390/jfmk7040115

**Published:** 2022-12-15

**Authors:** Corey A. Peacock, Duncan French, Gabriel J. Sanders, Anthony Ricci, Charles Stull, Jose Antonio

**Affiliations:** 1Department of Health and Human Performance, Fight Science Lab, Nova Southeastern University, Fort Lauderdale, FL 33328, USA; 2UFC Performance Institute, Las Vegas, NV 89118, USA; 3School of Health Sciences, Australia Catholic University, Melbourne, VIC 3065, Australia; 4School of Medical and Health Sciences, Edith Cowan University, Perth, WA 6027, Australia; 5Department of Kinesiology and Health, Northern Kentucky University, Highland Heights, KY 41009, USA

**Keywords:** MMA, weight cutting, weight classification

## Abstract

Previous research has demonstrated that professional mixed martial arts (MMA) athletes employ a variety of weight manipulation strategies to compete at given weight classes. Although there is much literature demonstrating weight manipulation methods, minimal research exists analyzing how much weight MMA athletes lose prior to the official weigh-in. Moreover, there is minimal research examining how much weight professional MMA athletes gain between the official weigh-in and competition. Therefore, the purpose of the current study was to analyze weight loss/regain in professional MMA athletes. Data collected from 616 professional MMA athletes (31.1 ± 4.0 yrs.; 177.1 ± 4.7 cm) competing for the Ultimate Fighting Championship (UFC) between 2020 and 2022 were used for the study. The athlete’s weight was obtained 72 h, 48 h, and 24 h prior to the official weigh-in, at the official weigh-in, and prior to competition. Random effects analysis was utilized to compare weight at a variety of time points between different weight classes. All statistics were analyzed, and significance was set at *p* ≤ 0.05. There is a significant (*p* ≤ 0.05) difference between weight classes and time points in professional MMA. MMA athletes decrease body weight significantly prior to the official weigh-in. MMA athletes increase body weight significantly between official weigh-in and competition. Based on these data, it appears that MMA athletes average a weight loss of nearly 7% within 72 h prior to the official weigh-in. The data also suggest that athletes gain nearly 10% of total weight between the official weigh-in and competition.

## 1. Introduction

While fight training for sport dates back thousands of years [1], the origins of Mixed Martial Arts (MMA) can be tied in the past to Ancient Greece in 648 BC, where participants would compete in pankration as the final event of the Olympics [2,3]. In the early 1990s, a resurgence in the popularity of striking and grappling sports emerged with the Ultimate Fighting Championship (UFC) [4,5]. Early MMA bouts were unregulated, and fighters exhibited a variety of fighting disciplines including boxing, Brazilian jujitsu (BJJ) and sumo-wrestling [4]. Due to increased popularity, MMA became lucrative and internationally sanctioned, and rules were modified to increase safety. These rules included the banning of hitting the spine or back of the head, pulling hair, and head butting [2,6,7]. Another of the most prominent rules included the creation of weight classes, as these were enforced to increase athlete safety and eliminate unfair advantages.

Like the weight class sports of wrestling and BJJ [8,9,10,11,12], weight “cutting” is a prevalent aspect in MMA. However, weight cutting is a more prevalent aspect in MMA than many other combat and grappling sports [13,14,15], as research demonstrates over 90% of MMA athletes use rapid weight loss techniques [14,16]. Research suggests an MMA athlete may lose up to 10% of their body mass prior to a weigh-in [8,9,10,11,12,17], and may regain most of that lost weight in the time between the weigh-in and the fight competition [18,19,20]. This in theory supports the idea that an athlete with a weight advantage may confer a competitive advantage in certain combat sports and contexts [10,12,19,21]. A recent combat sports analysis on exercise performance showed that some aspects of laboratory-based performance are unaffected or slightly impaired following certain weight loss strategies. Overall, they concluded that weight cutting does not affect overall exercise performance in a laboratory setting [10,12,19,21,22]. Weight can be lost rapidly through many modalities including heat, water loading, and fasting [17,23]. Despite the physical advantage, rapid weight loss can lead to dehydration [10,24] and many other negative health outcomes [23,25,26].

Weight classes in the UFC include strawweight (115 lbs/52.2 kg), flyweight (125 lbs/56.7 kg), bantamweight (135 lbs/61.2 kg), featherweight, (145 lbs/65.8 kg), lightweight (155 lbs/70.3 kg), welterweight (170 lbs/77.1 kg), middleweight (185 lbs/83.9 kg), light heavyweight (205 lbs/93.0 kg), and heavyweight (205–265 lbs/93.0–120.2 kg) [17]. Although previous research has explored weight cutting in MMA, there is currently a lack of research analyzing this at the professional level of the UFC. There is also a lack of literature comparing weight class practices during weight cutting. Therefore, the purpose of this study was to identify current weight cutting norms, to gain a better understanding of weight class practices. We hypothesize that weight cutting and weight gaining between classes will differ in UFC athletes.

## 2. Materials and Methods

A total of 616 professional MMA athletes (31.1 ± 4.0 yrs.; 177.1 ± 4.7 cm) competing for the UFC between 2020 and 2022 were used for the study. The athletes reported to the designated competition hotel 72 h prior to the official weigh-in (24–36 h prior to competition). While in the hotel, athletes provided self-reported weights at 72 h, 48 h, and 24 h prior to the official weigh-in. These self-reports were provided immediately following each weigh-in to UFC performance staff and were observed by the fighter’s team. All weights were obtained using a commission calibrated digital scale and were reported immediately to UFC performance staff. A weight was obtained utilizing the commission-managed beam or digital scale during the official weigh-ins and were reported publicly. The day following official weigh-ins, UFC athletes reported to the arena for competition where weight was obtained using a commission calibrated digital scale and recorded. Athletes who missed weight or did not officially weigh-in were excluded from the data set. It is worth noting that the designated hotel and arenas were in different states and countries, as there were 89 events held between 2020 and 2022. Athletes competing more than once between 2020 and 2022 were included as individual data points.

This study of de-identified data was approved by the University Institutional Review Board (IRB). All athletes provided consent to weigh-in and were deemed physically healthy for competition. Means and measures of variability were calculated for descriptive data. Normality was assessed and confirmed with the Shapiro–Wilk test and can be presented as mean ± standard deviation. A fixed effects ANOVA was used to analyze percentage changes by weight class. A random effects ANOVA was utilized to compare weight at a variety of time points and to compare different time points between different weight classes. Bonferroni post hoc analysis was utilized to further examine differences at each time point. All statistics were analyzed using Statistical Analysis Software (SPSS, Version 22.0, IBM INC., Armonk, NY, USA) and significance was set at *p* ≤ 0.05.

## 3. Results

### 3.1. Physical Characteristics

Physical characteristics including age and height were calculated for all athletes by weight class (Table 1).

### 3.2. Percent Weight Changes by Weight Class (Fixed Effects)

The fixed effects analyses revealed there was a significant main effect of weight class on percent weight changes (F = 1223.9, *p* < 0.001) throughout the 72 h prior to the weigh-in. Post hoc analysis revealed there were significant differences in the percent weight changes between all weight classes and the heavyweight class (*p* < 0.001) (Table 2). Percentage weight changes were derived from the difference in time-point relative to official weigh-in results. There were no other differences between any other weight classes (*p* = 0.433).

### 3.3. Percent Weight Changes by Time Point (Random Effects)

The random effects analyses revealed that there was a significant main effect of time point on percentage weight changes (F = 4836.9, *p* < 0.001). Post hoc analysis revealed there were significant differences between every time point (*p* < 0.001 for all, Table 3). Percentage weight reductions varied as follows: 72 h pre-weigh-in fighters lost −6.7% weight, 48 h pre-weigh-in fighters lost −5.7% weight, 24 h pre-weigh-in fighters lost −4.4% weight, and then, cumulatively, fighters gained 9.7% body weight after official weigh-ins.

### 3.4. Percent Weight Changes by Weight Class (Fixed Effects) for Each Time Point (Random Effects)

There was a significant weight class by time point interaction (F = 33.06, *p* < 0.001) for percentage weight change. Post hoc analysis revealed (Table 4) that percentage weight changes by fighters in various weight classes were significantly different at 72 h pre- (*p* = 0.016), 48 h pre- (*p* = 0.019), 24 h pre- (*p* = 0.029) and post-fight weigh-ins (*p* = 0.026).

## 4. Discussion

The results demonstrate a predominant aspect of weight cutting in professional UFC athletes apart from the heavyweight division. This demonstration is consistent with previous research analyzing other weight regulated sports [11,13,14,15,27,28]. The data set examines weight at 72 h pre, 48 h pre, 24 h pre, official weigh-in, and post-weigh-in competition weight. Consistent with previous research analyzing the frequency of rapid weight loss measures, the data demonstrate that nearly all athletes utilized rapid weight loss techniques for the official weigh-in (apart from the heavyweight division). Previous research suggests that an MMA athlete loses nearly 10% of their total body weight pre-weigh-in [8,9,10,11,12,17]. The current data demonstrate that UFC fighters will lose approximately 6.7% of their total body weight 72 h pre-weigh-in. The data also demonstrate that athletes will lose approximately 5.7% of their total body weight 48 h pre- weigh-in and approximately 4.4% of their total body weight in the final 24 h of weight cutting pre-weigh-in. Previous research has suggested that athletes will regain most of that weight that was lost prior to competition [18,19,20]. Moreover, it was suggested that athletes could gain nearly 5.5 kg of bodyweight prior to competition [29]. The current data demonstrate that UFC athletes lost nearly 6.7% of their total body weight in a 72 h period, however, the UFC athletes regained approximately 9.7% of their total body weight between official weigh-in and competition (24–36 h post). Previous rehydration research has shown that weight regained may or may not have an impact on overall winning and performance [27,30]. This may demonstrate that UFC athletes are more effective at weight cutting strategies when compared to other professional fighters utilized in previous research.

It was hypothesized that weight cutting percentages would vary between weight divisions. Based on the results, many differences existed between weight classes at specific time points. Most notably, the featherweight division demonstrated the most aggressive of all weight loss/regain data. This division lost approximately 8.2% in the final 72 h, while regaining nearly 12% of their total body weight. Many differences existed when comparing weight classes to the light heavyweight class. This class appeared to be the least aggressive, losing approximately 5.5% in the final 72 h while regaining nearly 7% of their total body weight. All weight classes were significantly different when compared to the heavyweight class, as most of these athletes do not cut weight due to the upper limit (205–265 lbs/93.0–120.2 kg).

Based on the current study, the results demonstrate that professional UFC fighters lose weight prior to the official weigh-in. Moreover, these athletes also gain weight to rehydrate, refuel, and ultimately compete at a weight heavier than that recorded at the official weigh-in. This is the first study to analyze professional UFC athletes at this volume regarding mean weight loss/regain for competition. Based on the results, these percentages may be applied to weight loss scheduling, may provide safe and effective weight loss strategies, and may improve weight regaining hydration prior to competition. While the current study is the first to assess UFC fighter weight cuts during fight week, it is not without limitations. First, the self-reported body mass values and the time duration is of limitation. The literature shows that less than or equal to seven days preceding competition captures the most significant degree of rapid weight loss [11,31]. Another limitation of the study include attire not being controlled for, but athletes were instructed to strip down to minimal clothing while stepping on the scale. Finally, baseline data was not collected, but would provide more detailed insight into the results. Future research should aim to assess multiple physiological variables including hydration, electrolytes, cardiovascular function, and neurological function. Further research is currently underway analyzing weight cutting strategies.

## Figures and Tables

**Table 1 jfmk-07-00115-t001:** Age and height for all athletes by weight class.

Weight Class	Sample (N)	Upper Weight Limit	Age (Years)			Height (cm)		
Strawweight	44	115 lbs (52.2 kg)	30.8	±	4.0	160.5	±	4.4
Flyweight	91	125 lbs (56.7 kg)	30.1	±	3.7	166.4	±	4.4
Bantamweight	113	135 lbs (61.2 kg)	30.8	±	4.0	170.7	±	4.1
Featherweight	77	145 lbs (65.8 kg)	30.6	±	3.8	175.3	±	4.8
Lightweight	79	155 lbs (70.3 kg)	30.8	±	4.2	177.3	±	4.6
Welterweight	73	170 lbs (77.1 kg)	31.9	±	4.1	181.6	±	5.8
Middleweight	72	185 lbs (83.9 kg)	30.2	±	3.8	184.4	±	4.1
Light Heavyweight	38	205 lbs (93.0 kg)	31.9	±	4.3	188.0	±	5.1
Heavyweight	29	265 lbs (120.2 kg)	32.5	±	4.3	190.0	±	4.8

Data are Means ± SD.

**Table 2 jfmk-07-00115-t002:** Percentage Weight Changes by Weight Class (Fixed Effects).

Percentage (%) Weight Change from 72 h Pre- to Post-Weigh-in Rehydration Weight				
Strawweight	−1.7	±	1.0	^a^
Flyweight	−1.9	±	1.1	^a^
Bantamweight	−2.1	±	1.0	^a^
Featherweight	−2.2	±	1.1	^a^
Lightweight	−2.0	±	0.9	^a^
Welterweight	−2.0	±	1.0	^a^
Middleweight	−2.0	±	1.0	^a^
Light heavyweight	−1.7	±	0.8	^a^
Heavyweight	−0.4	±	2.8	

Data are Means ± SD. ^a^ Significantly different from Heavyweight class, *p* < 0.001.

**Table 3 jfmk-07-00115-t003:** Percentage Weight Changes Each Weigh-In Day (Random Effects).

Percent (%) Weight Change				
72 h Before	−6.7	±	2.3	^a,b,d^
48 h Before	−5.7	±	2.1	^a,c,d^
24 h Before	−4.4	±	2.9	^b,c,d^
Post-Weigh-In	9.7	±	4.0	^a,b,c^

Data are Means ± SD. ^a^ Significantly different from 24 h pre-weigh-in, *p* < 0.001. ^b^ Significantly different from 48 h pre-weigh-in, *p* < 0.001. ^c^ Significantly different from 72 h pre-weigh-in, *p* < 0.001. ^d^ Significantly different from post-weigh-in weight, *p* < 0.001.

**Table 4 jfmk-07-00115-t004:** Percentage Weight Changes by Weight Class (Fixed Effects) for Each Weigh-in Day (Random Effects).

Percentage (%) Weight Change Pre- and Post- Official Weigh-in for Each Weigh-in Day																
	72 h before				48 h before				24 h before				Post-Weigh-In			
Strawweight *^,#,%,x^	6.5	±	1.9	^c,d^	5.4	±	1.8	^c,d,e,i^	4.1	±	1.7	^d,i^	9.3	±	3.2	^b,c,d,e,h,i^
Flyweight *^,#,%,x^	7.3	±	2.2	^h,i^	6.4	±	2.0	^h,i^	5.0	±	1.9	^i^	11.3	±	3.6	^a,h,i^
Bantamweight *^,#,%,x^	7.9	±	1.8	^a,g,h,i^	6.8	±	1.7	^a,h,i^	5.5	±	1.7	^h,i^	11.8	±	2.8	^a,g,h,i^
Featherweight *^,#,%,x^	8.2	±	1.8	^f,g,h,i^	7.0	±	1.9	^a,g,h,i^	5.9	±	1.9	^a,h,i^	12.3	±	2.9	^a,g,h,i^
Lightweight *^,#,%,x^	7.7	±	1.5	^h,i^	6.6	±	1.4	^a,h,i^	5.5	±	1.4	^h,i^	11.9	±	2.4	^a,g,h,i^
Welterweight *^,#,%,x^	7.1	±	1.3	^d,h,i^	6.1	±	1.4	^h,i^	5.1	±	1.4	^i^	10.4	±	3.1	^d,h,i^
Middleweight *^,#,%,x^	6.8	±	2.0	^c,d,h,i^	6.0	±	2.0	^d,h,i^	5.0	±	1.7	^i^	10.0	±	3.7	^c,d,e,g,h,i^
Light heavyweight *^,x^	5.6	±	1.5	^b,c,d,e,f,g,h,i^	4.6	±	1.5	^b,c,d,e,f,g,h,i^	3.7	±	1.9	^c,d,e,i^	6.9	±	3.2	^a,b,c,d,e,f,g,i^
Heavyweight *^,#,%,x^	2.8	±	2.9	^a,b,c,d,e,f,g,h^	2.2	±	2.6	^a,b,c,d,e,f,g,h^	0.1	±	10.3	^a,b,c,d,e,f,g,h^	3.1	±	3.3	^a,b,c,d,e,f,g,h^

Data are Means ± SD. ^a^ Significantly different from strawweight, *p* ≤ 0.026. ^b^ Significantly different from flyweight, *p* ≤ 0.026. ^c^ Significantly different from bantamweight, *p* ≤ 0.001. ^d^ Significantly different from featherweight, *p* ≤ 0.029. ^e^ Significantly different from lightweight, *p* < 0.007. ^f^ Significantly different from welterweight, *p* ≤ 0.007. ^g^ Significantly different from middleweight, *p* ≤ 0.05. ^h^ Significantly different from light heavyweight, *p* ≤ 0.05. ^i^ Significantly different from heavyweight, *p* < 0.001. * Significantly different from 72 h pre-weigh-in, *p* < 0.001. ^#^ Significantly different from 48 h pre-weigh-in, *p* ≤ 0.005. ^%^ Significantly different from 24 h pre-weigh-in, *p* ≤ 0.005. ^x^ Significantly different from post-weigh-in weight, *p* ≤ 0.006.

## Data Availability

Not Applicable.

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
