# Peer review of "Weight Loss and Competition Weight in Ultimate Fighting Championship (UFC) Athletes"

_jfmk, 2022, doi:10.3390/jfmk7040115_

Round 1

Reviewer 1 Report

Overview

The authors aimed to identify current weight-cutting norms in professional MMA athletes, to gain a better understanding of weight class practices. Also, they hypothesized that weight cutting and weight gaining between classes should differ in Ultimate Fighting Championship athletes. 

The topic is interesting. The manuscript is well written.

Specific comments

Introduction

The introduction is well-written. A good synthesis of the literature has been made. The gap to be filled is clear, and the purposes and the hypothesis to be tested have been well described.

Material and Methods

The methodology used is very simple and it is clearly explained.

To fix: Not all statistics used are described.

Results

Well written section.

Discussion

The conclusions are justified and the take home message is clear.

Author Response

Point 1: To fix: Not all statistics used are described.

Response 1: We updated to include the explanation of both the random effects and fixed effects analysis.  “Means and measures of variability were calculated for descriptive data. A fixed effects analysis was used to analyze percentage changes by weight class.  Finally random effects analysis was utilized to compare weight at a variety of time points and to compare differ-ent time points between different weight classes. All statistics were analyzed using Statis-tical Analysis Software (SPSS, Version 22.0, IBM INC., Armonk, NY) and significance was set at p ≤ 0.05.”

On Behalf of our entire team, I thank you for your time and very important review.  I believe with your suggestions, our work is much improved and now ready for publication. 

All the best,

Corey Peacock

Reviewer 2 Report

Thank you for the opportunity to review this paper. I have some comments that can hopefully help clarify and improve the reporting of the study. My main comments are on the lack of detail in the methods and statistics section, the presentation of results, and the critical research missing in the discussion section. 

Author Response

Please see the attachment-

Reviewer 3 Report

I kindly as the authors to review the manuscript to make it more sophisticated. As Its presentation does not seem high enough to be published in this journal. My point-by-point comments can be found below.

L18: should be “was to analyze”

L19-20: “616 professional MMA athletes (31.1 ± 4.0 yrs.; 177.1 ± 4.7 cm) 19 competing for the Ultimate Fighting Championship (UFC) between 2020 and 2022 were used for the 20 study.” Better to redefine the sentence as participants are not used but the data obtained from them can be used.

As far as I am concerned, the biggest limitation of the study was that athletes’ baseline body weight was not monitored (i.e., the weight before they applied weight loss). It would be useful to indicate how much weight they lost from baseline to the competition with the same time intervals.

L66: was to….

L73: How reliable can self-reported weight be? As a previous athlete with 18 y of weight cutting experience, I know that this is not a reliable way to prepare a scientific paper just basing on the knowledge provided from athletes.

L84: Please rewrite the statistics more explicitly, it is not understood what you did for what. I cannot understand the findings from this explanation (what are fixed and random effects) and thus I want to read the findings following you rewrite the statistics.

Results: Please calculate 95% CI for each variable and also if you applied repeated measures ANOVA, which you had to, please report effect sizes. I have never come across “/” while reporting statistics, why did you use them? You can report as p=0.435 or p>0.05

Table2: What do you mean by rehydration weight? Did you monitor athletes’ hydration status and become sure they were all rehydrated during this process? Literature shows that many combat sport athletes cannot rehydrate following weigh-in until competition.

 L147-148: Although you stated all athletes including heavyweights had a weight loss with p<0.001 significance (Table 2) you indicated heavyweights did not implement weight cutting: the data demonstrates nearly all athletes utilized rapid weight loss techniques for the official weigh-in (apart from the heavyweight division).

The discussion should be rewritten considering and comparing the findings in the literature with similar research approach (including other sports), examples:

https://doi.org/10.1080/00913847.2022.2026200

https://doi.org/10.1123/ijsnem.2020-0369

https://doi.org/10.1080/15438627.2021.1989435

Round 2

Reviewer 2 Report

I believe the paper has been improved. While other changes could improve the paper, I believe it is acceptable in its present form. 

I look forward to reading the other publications the group has planned. 

Reviewer 3 Report

The manuscript can be accepted in its current form.